# Somatic mutagenesis in satellite cells associates with human skeletal muscle aging

Irene Franco [1], Anna Johansson[2], Karl Olsson[3], Peter Vrtačnik[1], Pär Lundin[1,4], Hafdis T. Helgadottir[1], Malin Larsson[5], Gwladys Revêchon[1], Carla Bosia[6,7], Andrea Pagnani [6,7], Paolo Provero[8,9], Thomas Gustafsson[3], Helene Fischer[3] & Maria Eriksson[1]

Human aging is associated with a decline in skeletal muscle (SkM) function and a reduction in the number and activity of satellite cells (SCs), the resident stem cells. To study the connection between SC aging and muscle impairment, we analyze the whole genome of single SC clones of the leg muscle vastus lateralis from healthy individuals of different ages (21–78 years). We find an accumulation rate of 13 somatic mutations per genome per year, consistent with proliferation of SCs in the healthy adult muscle. SkM-expressed genes are protected from mutations, but aging results in an increase in mutations in exons and promoters, targeting genes involved in SC activity and muscle function. In agreement with SC mutations affecting the whole tissue, we detect a missense mutation in a SC propagating to the muscle. Our results suggest somatic mutagenesis in SCs as a driving force in the age-related decline of SkM function.

[1] Department of Biosciences and Nutrition, Center for Innovative Medicine, Karolinska Institutet, 14157 Huddinge, Sweden. [2] Science for Life Laboratory, Department of Cell and Molecular Biology, Uppsala University, 75237 Uppsala, Sweden. [3] Division of Clinical Physiology, Department of Laboratory Medicine, Karolinska Institutet, 14186 Huddinge, Sweden. [4] Science for Life Laboratory, Department of Biochemistry and Biophysics (DBB), Stockholm University, 10691 Stockholm, Sweden. [5] Science for Life Laboratory, Department of Physics, Chemistry and Biology, Linköping University, 58183 Linköping, Sweden. [6] Italian Institute for Genomic Medicine (IIGM), 10126 Turin, Italy. [7] Department of Applied Science and Technology, Politecnico di Torino, 10129 Turin, Italy. [8] Department of Molecular Biotechnology and Health Sciences, Molecular Biotechnology Center, 10126 Turin, Italy. [9] Center for Translational Genomics and Bioinformatics, San Raffaele Scientific Institute, 20132 Milan, Italy. Correspondence and requests for materials should be addressed to I.F. (email: irene.franco@ki.se) or to M.E. (email: Maria.Eriksson.2@ki.se)

Satellite cells (SCs) are a heterogeneous population of stem and progenitor cells that have been demonstrated to play a pivotal role in skeletal muscle (SkM) hypertrophy, regeneration, and remodeling[1,2]. The SCs are normally kept in a quiescent state and activated upon exposure to stimuli, such as exercise or SkM injury. When committed to myogenic differentiation, SCs proliferate further, fuse to existing SkM fibers, and contribute new nuclei to the growing and regenerating fibers[3]. Aged human SkMs show a decline in the number and proliferative potential of the SCs[4]. As a consequence, a dysfunctional SC compartment is envisaged as a major contributor to age-related defects, including reduced capacity to respond to hypertrophic stimuli such as exercise and impaired recovery from muscle disuse and injury[1,5,6]. Furthermore, SCs have been shown to contribute to differentiated fibers in non-injured muscles of adult sedentary animals[7,8]. The basal turnover of nuclei in adult fibers appears to be less crucial in the protection from sarcopenia[7], a progressive loss of SkM mass and function, which culminates in a highly disabling condition affecting up to 29% of the population aged 85 years[9]. Nonetheless, SCs play an essential role in limiting the occurrence of fibrosis in the SkM of mice affected by sarcopenia[7] and their function in the human pathology needs to be further characterized.

A well-known factor in the decline of stem cell function is the loss of genome integrity[10], for example, caused by the appearance of somatic mutations[11]. These modifications of the genome range from single-base changes (single-nucleotide variants (SNVs)) to insertions or deletions of a few bases (indels) to chromosomal rearrangements and occur during the whole life, starting from the first division of the embryo. In contrast to germline variants, somatic variants are not propagated to the whole individual but to a subpopulation of cells in the body, with the final consequence that adult human tissues are a mosaic of genetically different cells[12–14]. Moreover, somatic mutation burden increases during a lifetime[15–18] as a result of accumulating errors occurring either during cell division or because of environment-induced DNA damage. At present, nothing is known about somatic mutation burden in human SCs or SkM.

Here, we investigate the genetic changes that occur with aging in the genome of human adult SCs and use the results to elucidate mutational processes and SC replication rate occurring in vivo in adult human muscles. We assess the functional effects of somatic mutations on SC proliferation and differentiation and predict the global consequence on muscle aging and sarcopenia. Our analyses reveal an accumulation of 13 mutations per genome per year that results in a 2–3-fold higher mutation load in active genes and promoters in aged SCs. High mutation burden correlates with defective SC function. Overall, our work points to the accumulation of somatic mutations as an intrinsic factor contributing to impaired muscle function with aging.

## Results

**Increased somatic mutation burden in aged SCs**. We examined the somatic genetic variation in SCs from the leg muscle vastus lateralis of a group of young (21–24 years, $n = 3$) and old (64–78 years, $n = 4$) individuals by whole-genome sequencing (WGS). Because the sequencing of clonal populations is an effective strategy to study the genome of single cells[19,20], freshly isolated cells expressing the SC antigen CD56 were single cell plated and cultured for 17–20 divisions prior to DNA extraction (Fig. 1a). Even though the selection of a pure SC population from muscle biopsies requires the use of multiple markers besides CD56[21], the combined effect of our isolation protocol (Supplementary Fig. 1a) and culture conditions ensured the almost exclusive growth of satellite cell clones (SCCs, 57/58 tested colonies, Supplementary

Table 1). Two to five SCCs/biopsy and the relative SkM and blood bulk DNA were subjected to WGS with an average depth of 30× (Supplementary Table 2). The SC origin of the sequenced clones was tested with either a differentiation assay or quantitative PCR (qPCR) for the expression of the SC markers Pax7, MyoD, and myogenin. Twenty one out of 21 tested SCCs resulted positive (Supplementary Table 2 and Supplementary Fig. 1b and c).

Next, we analyzed somatic mutations in 29 SCCs. The identified somatic SNVs ranged between 209 and 1371 per SCC (Fig. 1b). Considering a calculated false negative rate of 0.41 (Supplementary Table 3), the actual number of somatic mutations per genome was estimated to range between 354 and 2323, which is consistent with studies in clonally expanded cells from human fibroblasts[19,22] and stem cells from the intestine and liver[16]. The pipeline for calling somatic variants was validated by analyzing two technical WGS replicates and showed a 94% and 99% validation rate of the somatic SNVs (Supplementary Table 4). In addition, WGS-independent validations of exonic SNVs selected from several clones ($n = 16$) showed a 100% validation rate (Supplementary Table 5). Somatic indels ranged between 8 and 109 per SCC (Fig. 1b and Supplementary Tables 2 and 6), with a validation rate of 84% and 98% in WGS of technical replicates (Supplementary Table 1D).

The somatic variants showed an age-dependent accumulation of mutations in the SC genomes (Fig. 1b; $P = 1.47 \times 10^{-4}$ for SNVs and $P = 2.77 \times 10^{-3}$ for indels), with a mutation rate of $13.1 \pm 2.7$ SNVs per genome per year during adult life (Fig. 1c). Somatic mutations have been suggested to arise in the quiescent state[13], but errors occurring during cell division are generally considered the major source of mutations[23,24]. If we only consider cell division-coupled mutations and use a mutation rate of 2.5 SNVs per genome per cell division[25,26], we can estimate a number of cell divisions per year in SCs of 5.24.

**Specific pattern of somatic mutations in adult SCs**. To obtain insight into the mutational processes occurring in SCs in vivo, we analyzed the profile of somatic substitution types. The somatic variant profile considerably differed from the germline spectrum assessed in the blood and muscle bulk of the same individuals (Supplementary Fig. 2a). In contrast, the mutational profiles of young and old SCCs were similar (Supplementary Fig. 2b). The substitution classification was further refined by including the 5′ and 3′ sequence context of each mutated nucleotide to obtain 96 mutation classes. Following established methods[16,27], we extracted 3 SC-specific signatures (Supplementary Fig. 3a) that represent the distinct mutation processes occurring during SC aging. Principal component analysis (PCA) of extracted signatures for our data and published data from human fibroblasts[19] showed separate clusters for the two distinct cell types. The pure cluster including all 29 SSCs confirmed the SC origin for all clones, including the 8 clones where no material was available for validation (Supplementary Fig. 3b). We also studied how previously established signatures (cancer signatures: http://cancer.sanger.ac.uk/cosmic/signatures) contributed to the SC mutation profiles (Supplementary Fig. 4 and Fig. 1d–g). SCC mutations can be principally attributed to signature 1, which is associated with the deamination of 5-methylcytosine, signature 5 and signature 8 (Supplementary Fig. 2a and Fig. 1d). Signature 5 contributed the highest number of mutations and the age-related increase was higher for signature 5 than for the other signatures (Fig. 1f, g). Interestingly, old SCCs presented a higher fraction of signature 5 mutations compared to young SCCs (Fig. 1e). This indicates that signature 1 and 8 mutations steadily accumulated during a lifetime, while the accumulation rate of signature 5 specifically increased between 20 and 70 years of age. Signature 18, which was

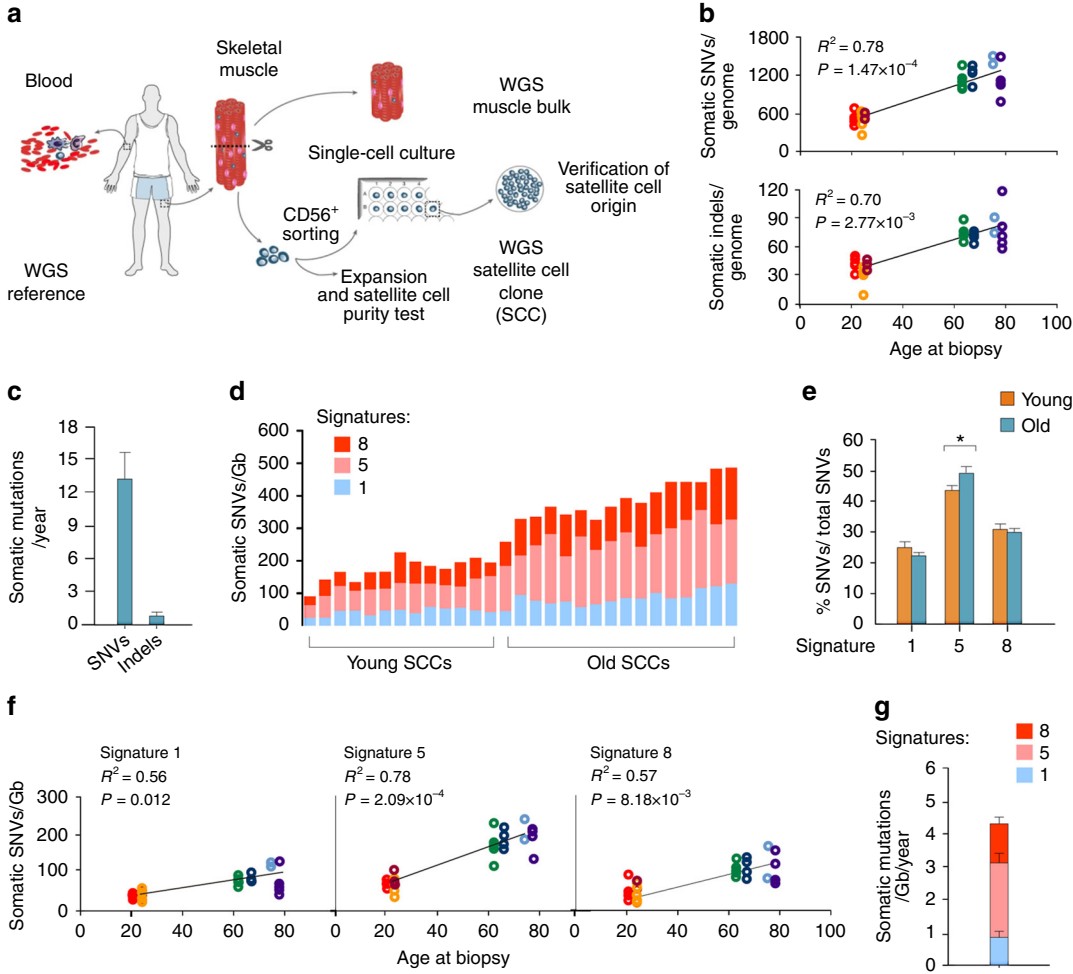

**Fig. 1** Age-related increase of somatic mutations in satellite cells. **a** Experimental strategy for somatic variant detection with whole-genome sequencing (WGS). The reference sequence was obtained from the blood of every individual. The muscle vastus lateralis was sequenced as a population (muscle bulk) or used to isolate CD56+ cells. These were either tested to verify the percentage of satellite cells or single cell plated and sequenced as a clonal population (SCCs) in order to detect cell-specific somatic mutations. The satellite cell origin of sequenced SCCs was tested in all clones that produced a sufficient number of cells (21/29) **b** Total number of somatic single-nucleotide variants (SNVs) and insertion/deletions (indels) per SCC normalized to the percentage of autosomes covered by the sequencing. The linear fit of mutation numbers and age is shown. *P*-values are calculated using a robust mixed model. Different colors correspond to the donors. **c** Average number of SNVs and indels accumulated per year by the SCCs according to the linear fit shown in **b**. **d** Absolute contribution of signatures 1, 5, and 8 to the mutation catalogue of each SCC. **e** Average percentage of SNVs/total SNVs for the most relevant signatures in young and old SCCs. *$P < 0.05$ two-sided *t*-test. **f** Linear fit of cancer signatures 1, 5, and 8 with the age of the SCCs. *P*-values were calculated with a robust mixed model with Bonferroni correction for multiple testing of different signatures ($n = 3$). **g** Yearly increase in the selected cancer signatures obtained by the linear fits in **f** was plotted as a proportion of the total number of SNVs/Gb accumulating every year in the SCCs

previously associated with culture-induced somatic mutations[16], was almost absent in SCCs cultured with our standard protocol but increased in a control subset of SCCs that underwent an additional 50 days of in vitro expansion between the isolation and single-cell plating (Supplementary Fig. 4b). The control subset also presented a higher fraction of C>A conversions (Supplementary Fig. 2b), which is consistent with culture-induced mutation processes[16]. These results confirm that our strategy for calling somatic variants efficiently excluded mutations that occurred during in vitro culture.

**Mutation accumulation in SkM-expressed exons and promoters**. We analyzed the somatic mutation distribution across the genome. Plotting the variants according to either their chromosome position (Supplementary Fig. 5a) or annotation

(Table 1) did not show a specific pattern or any age-related differences other than the increased mutation burden associated with age (Fig. 2a). However, in both young and old SCCs, the mutation frequency was lower than expected in exons and regulatory regions (Fig. 2b, c). The under representation was more pronounced in the exons expressed in the SkM than in all the exons ($P = 4.55 \times 10^{-6}$) (Fig. 2b), consistent with a specific protection of functional regions. The protection extends to the introns of the same genes (Supplementary Fig. 5b), suggesting that it is due to the activity of the transcription-coupled repair. Interestingly, mutation depletion in exons and regulatory regions of young SCCs was stronger than in old SCCs (Fig. 2b, SkM-expressed exons; and Fig. 2c, promoters), while introns were equally protected from mutations in young and old SCCs (Supplementary Fig. 5b). These results indicate that the regions of the genome involved in the production of the proteome were

**Table 1 Average distribution of the annotated variants in young and old SCCs**

|  | Young, 23 ± 0.9 years | Old, 71.2 ± 4 years |
|---|---|---|
|  | % of SNVs | % of SNVs |
| Exon[a] | 0.9 ± 0.1 | 1.0 ± 0.1 |
| 3′ UTR | 0.3 ± 0.1 | 0.3 ± 0.0 |
| 5′ UTR | 0.0 ± 0.0 | 0.0 ± 0.0 |
| Splice region[b] | 0.1 ± 0.0 | 0.1 ± 0.0 |
| Intron | 40.0 ± 0.4 | 39.5 ± 0.4 |
| Upstream gene | 6.3 ± 0.4 | 6.4 ± 0.3 |
| Downstream gene | 6.3 ± 0.4 | 6.7 ± 0.4 |
| Regulatory region[c] | 2.6 ± 0.3 | 2.8 ± 0.2 |
| Non-coding transcript exon | 0.7 ± 0.1 | 0.6 ± 0.1 |
| Intergenic | 42.8 ± 0.6 | 42.6 ± 0.5 |

Percentage of mutations for each genomic feature reported as the mean ± standard error for young and old SCCs. Variants were annotated with Variant Effector Predictor (VEP)
[a]Categories reflects Ensemble SO terms, except for exon: which includes stop gained, missense, synonymous variants
[b]Categories reflects Ensemble SO terms, except for splice region: which includes splice region, splice donor, and splice acceptor variants
[c]Categories reflects Ensemble SO terms, except for regulatory region: which includes regulatory region and TF-binding site variants

subjected to a specifically higher accumulation of mutations in aged SCs. In addition, old SCCs showed a higher ratio of non-synonymous (potentially impairing the function of the gene) over synonymous (neutral) mutations (Fig. 2d), meaning a higher number of potentially deleterious mutations compared to young SCCs. Overall, these results point to a specific degeneration of active genes and promoters with aging.

**Somatic mutations in active genes during SC differentiation.** To obtain insight into the mutation load in specific genes ensuing differentiation and proliferation in SCs, we mapped our muta-tions to expression data generated by the FANTOM5 project[28]. These data provide a map of the exons, promoters, and enhancers that are active during the in vitro differentiation of myoblasts to myotubes (Supplementary Fig. 6), a process that mimics in vivo SC activation and fusion to differentiated fibers. We found that old SCCs carried an average of 11.5, 2.25, and 7.37 mutations per genome in the enhancers, promoters, and exons, respectively (Fig. 3a). These values were 2.4-fold higher than those in young SCCs. Notably, in the exons and promoters that were specifically expressed upon exposure to differentiation stimuli, the increase was more pronounced (fold increase 3.06 and 3.26 in exons and promoters, respectively), indicating that the chances of impairing the expression of a gene that is fundamental for SC differentiation strongly increased with age.

**Mutation load coincides with functional defects in old SCs.** In accordance with the possibility that somatic mutations impaired the function of important genes for SC differentiation, the number of SCCs able to differentiate into myotube structures inversely correlated with the average number of mutations per biopsy (Fig. 3b). We ruled out that the lack of myotube formation was caused by the non-myogenic origin of the SCCs by assessing the expression of MyoD in a subset of clones that did not respond to the differentiation treatment ($n = 14$). In addition, the colony-forming ability and long-lasting proliferative potential inversely correlated with the average number of mutations per genome in the SCs derived from the seven biopsies included in this study (Fig. 3c), supporting the hypothesis that higher numbers of mutations impaired cell proliferation.

Furthermore, we analyzed the proliferative capacity of individual SCCs that underwent WGS. We noticed that the SCCs that showed a higher fraction of senescent and un-proliferative cells required longer times to reach the 30,000–100,000 cells that are necessary for DNA extraction. Thus we used the time in culture as a measure of healthy proliferation and investigated whether the number of somatic mutations influenced this parameter. No correlation was found for young SCCs (Fig. 3d), which is consistent with the low mutation burden and the lack of an effect on proliferation. In contrast, in the old SCCs the proliferation time gradually increased with increased mutation burden (Fig. 3d). In order to define a mutation threshold that old SCCs can tolerate without showing proliferation defects, we considered as a range of healthy proliferation the average ± standard deviation of the days in culture of the young SCCs. The intersection of the upper limit of this range with the linear regression curve for old SCCs defined the mutation threshold and was calculated as 1250 mutations per clone. Even though this number is an underestimation, due to the stringent strategy for calling somatic mutations that allowed a false negative rate of 0.41 (Supplementary Table 3), it was useful to define 2 subsets of old SCCs. The population that carried < 1250 mutations was most often placed in the range of normal proliferation (healthy old SCCs, $n = 9$), while the majority of SCCs carrying > 1250 mutations showed abnormally long times in culture (diseased old SCCs, $n = 7$). This is an indication that a high number of somatic mutations affect the cell function. To understand the functional effects of the mutations, the variants were given a lethality score using CADD (Combined Annotation Dependent Depletion)[29]. For every clone, a global effect of somatic mutations was defined by the sum of CADD scores. The global CADD score was calculated exclusively for the mutations that were harbored in SC-expressed regions (according to FANTOM basal data shown in Fig. 3a) (Fig. 3e) or including all the mutations of each clone (Fig. 3f). Interestingly, the lethality score derived from mutations in expressed regions was not significantly different for healthy and diseased SCCs, while the score based on the complete set of mutations was higher for diseased SCCs (Fig. 3f). All together, these data provide a functional correlation between the extent of the somatic mutation burden and the impairment of SC activity.

**Somatic mutations in muscle functional genes in old SCs.** A mutation burden that leaves the SC proliferative capacity intact can also affect the muscle function by propagating to the genome of the SC progeny (for example, differentiated fibers). Overall, young clones presented an average of 5 exonic mutations per SCC, compared to 11 in old SCCs (Fig. 4a). Hence, we explored multiple sets of genes known to drive muscle changes, such as the response to training and aging. Genes differentially expressed after muscle training or participating in pathways involved in the response to training[5] (Fig. 4b and Supplementary Tables 7–9) showed a 5–6-fold increased mutation load in aged compared to young genomes. The same increase was observed for genes differentially expressed with age. Higher mutation loads in these genes increases the chances of defects in the muscle ability to adapt to training and aging.

To determine whether the genes that are linked to muscle diseases are more prone to somatic mutations, we mapped our somatic SCC mutations to 173 genes that are known to harbor pathogenic germline mutations (Fig. 4b). The results identified 4 genes (PLEC, SYNE2, TTN, and HSPG2) with somatic mutations in exons, but the average number of mutations in the entire gene set was not higher than expected when considering exons, introns, and flanking regions ($P = 0.1$ and $P = 0.45$ for young and

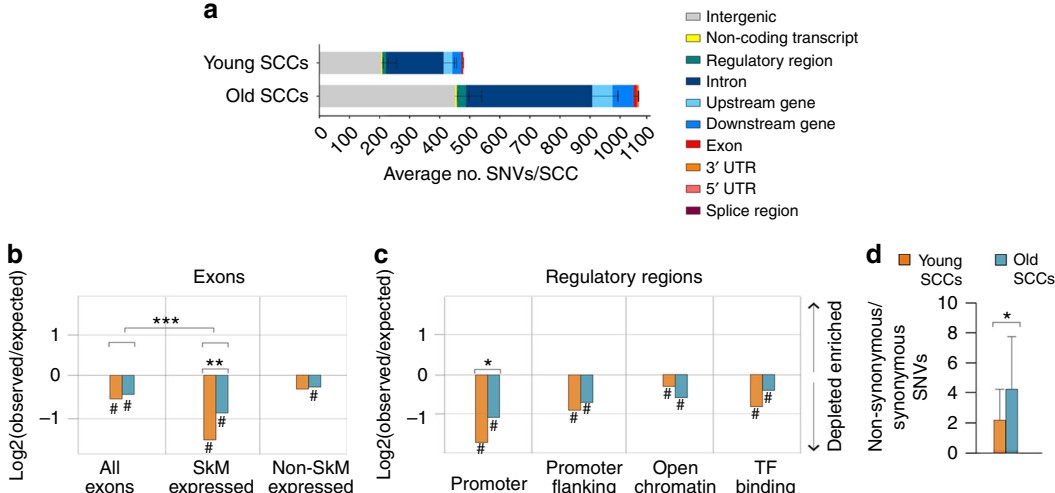

**Fig. 2** Non-random genomic distribution of somatic SNVs in the genome of SCCs. **a** SNV distribution in annotation classes according to the Ensembl Variant Effector Predictor. The graph represents the absolute numbers of SNVs per clone as the mean ± standard error for young and old SCCs. **b, c** Depletion of somatic SNVs in the indicated genomic regions in young and old SCCs. The list of skeletal muscle (SkM) expressed and non-expressed genes was obtained from the Human Protein Atlas (http://proteinatlas.com). The $\log_2$ ratio of the number of observed and expected point mutations indicates the effect size of the enrichment or depletion in each region. #$P < 0.05$, one-sided binomial test. **$P < 0.01$ ***$P < 0.001$ two-sided $t$-test of specified log2 ratios. **d** Non-synonymous/synonymous ratio of coding SNVs for young ($n = 10$) and old ($n = 14$) SCCs. *$P < 0.05$ Mann–Whitney non-parametric test

old SCCs, respectively. See Supplementary Table 10 and Supplementary Note 1 for a complete list of mutations and muscle disease genes).

**Propagation of SC mutations in SkM.** Finally, to analyze whether the potentially pathogenic somatic mutations that were detected in the SCs were actually propagated to the SkM, we designed assays for the rare event detection of a selection of mutations (Supplementary Table 11). As expected, these mutations were detected at an allele frequency close to 50% (48.7–49.5%) in the SCC from which they were originally discovered. Five out of the six variants were not detected in the corresponding muscle biopsy, which is consistent with their presence only in the sequenced SC or a fraction of cells below the limit of detection of our assays (Fig. 4c, and Supplementary Table 12). However, one variant, *HSPG2* c.7825C>T; p.R2609W, was present in 1.3% of the alleles of gDNA and 1.9% of the RNA transcripts from the corresponding muscle biopsy (Fig. 4c and Supplementary Fig. 7, Supplementary Table 12). The detected mutation in SkM was likely not derived exclusively from the SCs in the tissue, as the mutation was not identified in a population of CD56+ cells or in additional SCCs from the donor (Supplementary Table 12). Hence, this result supports the possibility that somatic mutations detected in SCs are propagated to the adult tissue, thereby increasing the negative effects of pathogenic variants.

## Discussion

Here, we have shown that SCs of the leg muscle vastus lateralis accumulate somatic mutations during adulthood at a rate of 13 mutations per genome per year. Somatic mutation data have been used as tracks on the genome to provide information about the mutational processes and stem cell activities that characterize the homeostasis of the uninjured muscle during human aging.

First, we provided evidence that the somatic mutations detected with our strategy were true variants that occurred during

in vivo aging and not artifacts derived from either the WGS sequencing or the in vitro cell culture. We tested the false positive rate with multiple validation methods to quantify the amount of errors due to the strategy for calling somatic variants. More than 94% of SNVs were validated, thus confirming our strategy for discovering somatic variants. The detection of indels is technically challenging[30], and the analysis of somatic indels has been seldom addressed. However, we were able to compile lists of somatic indels with a validation rate of 84–94% and add these mutations to the catalog of somatic mutations of adult SCs. To limit the inclusion of SNVs and indels that had arisen during the culture of the clone, our stringent filtering allowed only variants showing an allele frequency between 0.4 and 0.6. In fact, variants that occurred during culture would be present only in a subpopulation of the cells. Unless mutations were non-neutral, a mutation must have occurred during the first cell division to yield an allele frequency of 0.25. To validate our strategy experimentally, we analyzed the mutation pattern in a set of "positive control" samples for culture-induced variants, e.g., SCCs obtained from cells that stayed 50 days in culture before the cloning and subsequently cultured as regular SCCs. These long-culture SCCs showed features previously associated with cell culture-induced mutations, such as a high fraction of C>A conversions and the cancer signature 18[16]. Conversely, these features were almost absent in regular clones, thus confirming that our strategy for calling somatic variants efficiently excludes mutations that occurred during in vitro culture and provides a faithful representation of the genome of the SC at the moment of the isolation from the muscle.

A drawback of the strategy was the high false negative rate (0.41), mainly attributable to the stringent filtering on the allele frequency, which allowed for the discovery of only a portion of the true variants. This has implications for all our findings. For example, we calculated an age-related increase of 13.1 somatic mutations per genome per year and roughly estimated the rate of cell divisions per year in the SCs to 5.24. Both numbers are underestimated. However, these findings provide evidence in favor of a basal proliferation of SCs in adult muscles not

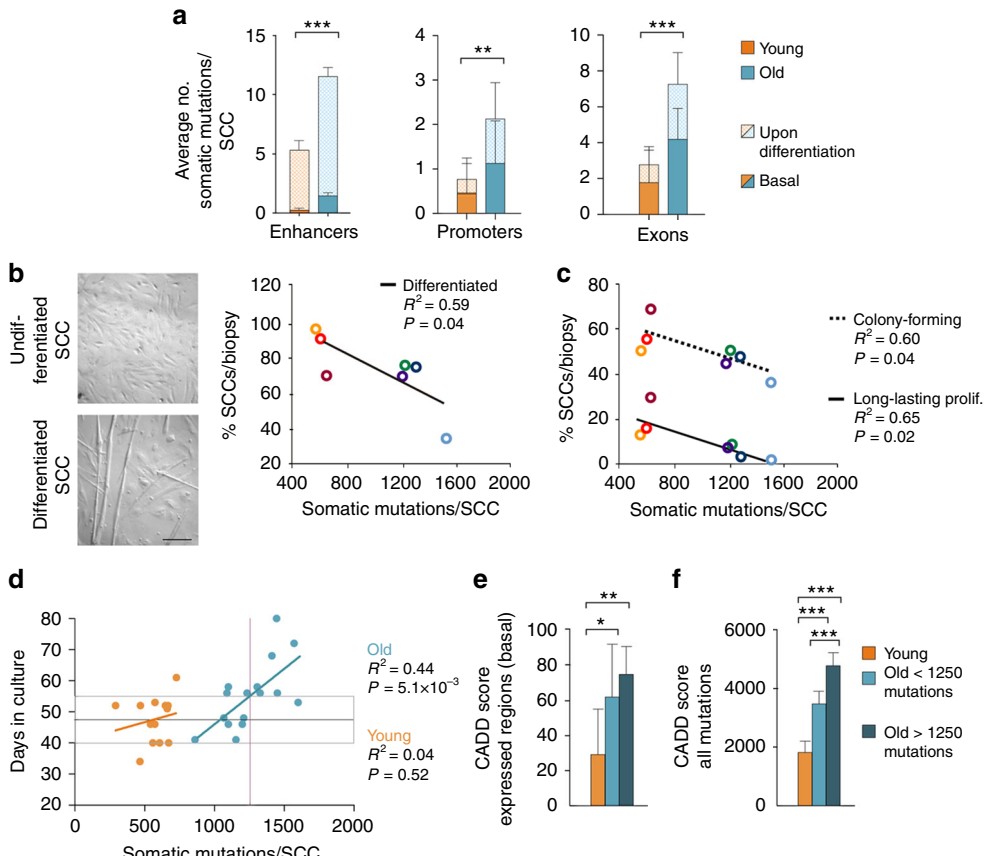

**Fig. 3** Mutation burden affects the satellite cell function. **a** Average number of mutations (SNVs + indels) in the specified genomic regions of young and old clones. The enhancers, promoters, and exons include only those regions that are actively transcribed during the differentiation of myoblasts to myotubes according to the FANTOM5 data[28]. Basal: regions expressed in SCs before induction of differentiation; upon differentiation: expressed only after differentiation treatment (see Methods). **P < 0.01, ***P < 0.001 two-sided t-test of average number of mutations in expressed regions (basal + upon differentiation) per clone. **b** An example of SCC cells in undifferentiated (growing medium 20% serum, top) and differentiated (1% serum, bottom) state and quantification of the percentage of SCCs that are able to differentiate after 5 days. Scale bar, 500 μM. A linear fit of the percentage of differentiated SCCs and the average number of somatic mutations per biopsy is shown. A subset of SCCs that failed differentiation was tested for MyoD expression to exclude their non-myogenic origin (CES3 n = 4, CES6 n = 3, CES7 n = 2, CES8 N = 5). **c** For every biopsy (young n = 3, old n = 4 individuals), at least 96 SCs were plated and the percentage of cells able to form a colony after 2 weeks in culture (colony forming) and the percentage of cells able to grow up to 30,000–100,000 cells (long-lasting proliferative potential) were registered. Linear fit of proliferation parameters with the average number of mutations detected in SCCs from each of the seven muscle biopsies included in the study is shown. **d** Linear fit between the time required for a single SC to proliferate to 30,000–100,000 cells (~17 divisions) and the number of somatic mutations/SCC. No correlation was found for young SCCs (orange line), in line with mutation burden compatible with healthy proliferation in young cells. The normal time in culture for a healthy clone (boxed area) was thus defined as the average (gray line) ± standard deviation of time in culture of young clones (n = 13). The intersection between the higher border of the boxed area and the linear regression curve for old SCCs (blue line) defined the maximum threshold of mutations above which SC proliferation is compromised (purple line, 1250 mutations). **e,f** The pathogenic effect of mutations, predicted using CADD (Combined Annotation Dependent Depletion). The global score was calculated as the sum of CADD PHRED-like scores of mutations found in either basally transcribed regions described in Fig. 3a (**e**) or in the whole genome (**f**). Bars show the average value per SCC in the young, healthy old (carrying < 1250 mutations, n = 9) or diseased old (> 1250, n = 7) groups

challenged by an injury or a disease. In particular, SCs are considered to stay mainly in a quiescent, non-proliferative state under basal conditions and maintain a very low metabolic rate to protect against DNA damage[4]. The proliferation of adult SCs in a muscle that does not need to heal from a trauma is debated. Some evidence has been provided in mice[7,8], but a direct demonstration in the human muscle was still lacking. The yearly mutation rate calculated from our data points to a sustained proliferation rate. However, we cannot determine whether the mutation accumulation is due to a steady (basal) proliferation or due to a few bursts occurring occasionally over decades. Moreover, we cannot establish whether the proliferation rate is attributable to a small fraction of dividing SCs, as was shown in the muscle of adult mice[31].

Beyond the quantitative data, an accurate analysis of the mutation type, pattern, and location allowed the understanding of mutation processes occurring in vivo in the adult human muscle. Mutagenic events and their effects on the genome have been collected in a catalog of mutational signatures (cancer signatures: http://cancer.sanger.ac.uk/cosmic/signatures). Of these signatures, 1 and 5 have previously been reported to increase with age in cancer and non-cancer tissues[15,16]. These signatures were the main signatures found in adult SCCs, together with signature 8. All three signatures progressively accumulated during aging. However, signature 5 showed the highest accumulation rate and a specific increase in adulthood compared to the period preceding 20 years of age. Unfortunately, the etiology of this signature is unknown.

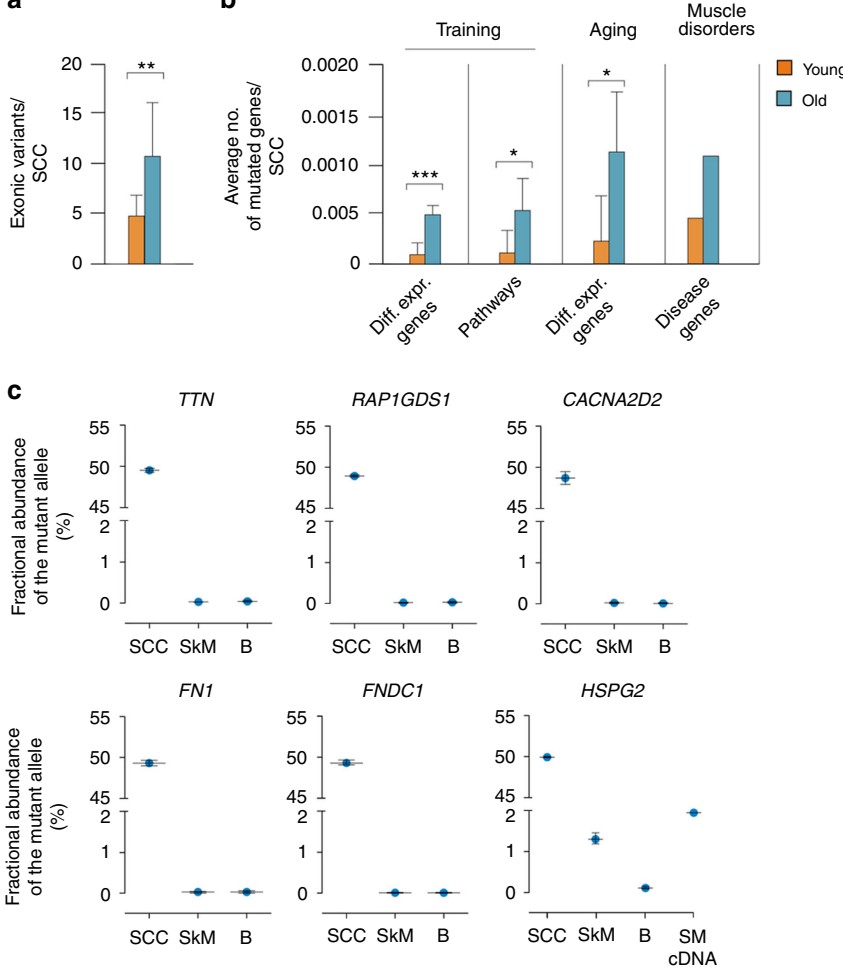

**Fig. 4** Propagation of SC mutations to the SkM. **a** Average number of exonic variants (SNVs + indels) in young and old clones. **P < 0.01 two-sided *t*-test. **b** Average number of genes harboring a mutation in young and old SCCs for different gene sets: differentially expressed after training (first); involved in pathways modulating the response to training[5] (second); differentially expressed with age (third); responsible for genetic muscular diseases (fourth graph). Gene lists were obtained from published data and public databases (see Supplementary Table 2). For muscle disease genes, see Supplementary Table 3. *P < 0.05, ***P < 0.001 two-sided *t*-test. **c** Abundance of mutated alleles of the indicated genes in the DNA of the clone from which the variant was originally discovered, the muscle, and the blood bulk of the related individual. The number of DNA molecules was detected by ddPCR and expressed as fractional abundance of all detected alleles (reference and mutant). For the *HSPG2* mutation c.7825C>T, the fractional abundance was also measured in muscle cDNA to assess transcription of the mutant allele. SCC satellite cell clone, SkM skeletal muscle, B blood

The analysis of the genome distribution of somatic variants indicated that several genetic regions harbor a number of mutations that is lower than expected by random distribution. The strongest protection from mutagenesis can be seen in the regulatory regions and in the SkM-expressed genes. Since these are transcribed regions, a likely mechanism involved in their protection from mutations is the transcription-coupled repair, a process in charge of repairing errors recognized by the Polymerase-2 on the template DNA during transcription[32]. Interestingly, both introns and exons are depleted of mutations, and this is in line with the fact that they are actively transcribed. However, only exons show a difference between young and old SCCs, with the young exons being more efficiently protected from mutations. This difference can also be observed in regulatory regions and can be attributable to either a progressive accumulation of mutations due to errors in the transcription-coupled repair or an age-related decay of the mechanisms in charge of specifically repairing the exons[33]. Another hypothesis is linked to the selective pressure. A specific feature of exons is that they are subjected to natural selection, e.g., non-functional alleles are rarely found in a population of cells, because cells that carry a defective gene are outcompeted by cells with functional alleles and disappear. A measure of the selective pressure acting on exons is the ratio of non-synonymous (potentially impairing the function of the gene) over synonymous (neutral) mutations. In agreement with a reduction of the selective pressure with aging, old SCCs showed an increased non-synonymous to synonymous mutation ratio compared to young SCCs. We hypothesize that the high mutation burden makes the majority of old SCs unfit for propagation and removes them from the environment. As a consequence, the strongly reduced number of functional cells is subjected to a milder selective pressure, in line with our data showing a lower number of proliferating SCs isolated from old muscles. Alternatively, the aged SC population may favor the clonal dominance of SCs that carry advantageous mutations, as seen for the hematopoietic and intestinal stem cells[34].

A specific accumulation of mutations in transcribed regions was also observed with a different approach, i.e., mapping somatic mutations to the transcriptome of SCs obtained from the CAGE (cap analysis gene expression) RNA seq data of the FANTOM

project[28]. These data provide a map of the exons, promoters, and enhancers that are active in cultured SCs and represent a set of genes important for SC maintenance and proliferation. Moreover, the regions expressed in response to differentiating stimuli were registered, giving an indication of the enhancers and genes important for SC differentiation and fusion to myotubes. In all of these regions, old SCCs carried a number of somatic mutations that was 2–3-fold higher compared to young cells.

We wondered whether the higher mutation burden could cause an impairment of the SC functions, e.g., proliferation and differentiation. Alterations in the stem cell function during aging have been ascribed to deregulation of both extrinsic and intrinsic factors[35]. Extrinsic factors include molecules systemically delivered to the muscle or secreted by the local niche in the microenvironment where the SCs reside. It has been demonstrated that the SC environment changes over time and influences the SC self-renewal and capacity for tissue regeneration[36]. On the other hand, the age-related reduction in the number of SCs that are able to repopulate the niche and differentiate into muscle fiberrs has been demonstrated to also rely on cell-intrinsic factors, such as the activation of signaling pathways[37,38] and epigenetic mechanisms[39]. With our model, we could explore how an intrinsic factor, i.e., genetic alterations, affects the SC activity, providing unique genome-wide data and direct assessment of the ex vivo performance of SCs of different ages. First, we found a negative correlation between the number of cells able to proliferate and differentiate and the average mutation burden in our biopsies. However, these correlations were based on data from the SC populations. To analyze the specific proliferation ability linked to each sequenced genome, we used data on the time required to complete the clonal culture. We found that the longer the time required for the colony to expand in culture, the higher the number of somatic mutations in the genome of the founder cell. This was true only for old SCCs, while no correlation was found for young SCCs. Overall, these results suggest that there is a threshold of mutations above which SC proliferation is impaired. Young SCCs showed a mutation burden lower than this threshold and proliferated normally. Conversely, old SCCs that carried a number of mutations above the threshold generally completed the clonal culture in abnormally long time. Similar results were obtained when we attributed a lethality score to every somatic mutation using CADD[29]. Old SCCs with normal proliferative rate and old SCCs with impaired proliferative capacity can be grouped and distinguished on the basis of the mutation lethality score. Interestingly, the lethality score that was only based on mutations in expressed regions could not distinguish normally and abnormally proliferating old SCCs. Hence, the effect of somatic mutations on cell fitness was dependent on the whole mutation burden, not only on expressed regions. In summary, our data on the proliferation of sequenced SCCs support the mutation burden as an important intrinsic factor regulating SC function during aging.

Besides affecting the SC activity per se, somatic mutation burden can be propagated to the mature tissue if SCs fuse to differentiated fibers. Therefore, we assessed whether mutations found in the SCs could be detected in the tissue of origin. To this end, we tested six SNVs affecting exons and introns of muscle-expressed genes and that showed high CADD scores. We found that a missense SNV in the *HSPG2* gene was present in 1.3% of the alleles, which relates to 2.6% of the cells of the muscle of the old individual from which the SC was derived. We also found that this mutation was expressed at detectable levels. This *HSPG2* variant has not previously been reported as pathogenic. However, it is harbored in a conserved region (PhastCons = 1) and a missense variant that changes the same amino acid (p.R2609Q)

was found in two different genetic diseases, dyssegmental dysplasia (OMIM 224410) and Schwartz-Jampel syndrome type 1 (OMIM 255800). Thus these results show that variants found in single SCCs can be found at detectable levels in the muscle and, if pathogenic, hamper the function of the whole tissue. In line with this view, we used the somatic variants found in SCCs to predict possible deleterious effects of somatic mutations in aged SkMs. We analyzed the mutation load in a number of gene sets that regulate muscle function such as genes differentially expressed after muscle training, pathways involved in the response to training, especially those that deteriorates with aging[5], and genes differentially expressed with age. Our results showed a mutation load increase of 5–6-fold with increased age and suggested a link between the increased mutation burden in old SCs and the defective response of aged muscles to training.

In summary, we have shown that SCs of the vastus lateralis muscle accumulate somatic mutations during life and provided evidence that intrinsic factors, such as the loss of genome integrity, contribute to aging of the SkM. Transcribed regions were specifically affected, and the mutation burden correlated with the capacity of SCs to proliferate and differentiate. Moreover, SCs could propagate their genomic defects to differentiated fibers and these defects might explain the decrease of muscle mass and response to stimuli observed in sarcopenia. Additional studies are required to assess similarities and differences in the age-related accumulation of mutations in muscles other than the vastus lateralis.

## Methods

**Clonal expansion of human SCs**. Human SkM biopsies were obtained from healthy volunteers according to Ethical Permit EPN 2015/847-31/1, the Stockholm regional ethical review board. Informed consent was obtained from all donors. In total, 50–100 mg of tissue from the muscle vastus lateralis were obtained using a percutaneous needle biopsy technique[40]. A part of the tissue was frozen, while ~40 mg were freshly digested using 5 ml Trypl-LX in two rounds of 20 min incubations at 37 °C, 5% $CO_2$ and gentle agitation. Cells were pre-plated in the growth medium in a 10-cm cell culture dish (Dulbecco's modified Eagle's medium (DMEM) F12 containing 1% ABAM and 20% fetal bovine serum (FBS)) for 30 min at 37 °C and 5% $CO_2$. The non-attached cells were transferred to a 6-cm cell culture dish coated with collagen I (5 μg/cm$^2$ of Collagen I bovine protein, Gibco #A10644-01, following the "thin coating procedure"). The cells were left to adhere for 48 h. Then the cells were detached and stained for CD56-PE (clone MY31, BD #347747, dilution 1:200). Using a BD FACSAria™ Mu cell sorter (BD Biosciences, USA), the cells were sorted based on their size and strong positivity for the antigen (see Supplementary Fig. 1) and were single cell plated in at least one 96-multiwell collagen-coated plate per biopsy. The cells were grown in a conditioned medium, i.e., growth medium that was collected every 48 h from a culture of feeder SCs (40–60% cell confluency), filtered through a 2-μm-wide pore filter and supplemented with FBS 1/10 of the volume. Colonies were counted after 14 days and scored as positive when at least 16 cells were visible in the well. Approximately 21 days after plating, the colonies were moved to 24-multiwell plates. Depending on the cell confluency, the colonies were then moved to six-multiwell plates. The new plates were collagen coated but growth medium instead of conditioned medium was provided. After an average of 52.9 ± 2.2 days in culture, the colonies were confluent and used for the DNA extraction.

**DNA extraction**. DNA was extracted from the confluent wells of the six-multiwell plate using the Gentra Puregen Kit, Qiagen. DNA was extracted from 30 mg of muscle biopsy using the Gentra Puregen Kit, supplemented with a lysis buffer containing Proteinase K as recommended by the supplier. DNA was extracted from 3 ml of total blood that was collected in EDTA as recommended by the instructions of the Gentra Puregen Blood Kit.

**Whole-genome sequencing**. The library preparation and sequencing were carried out at NGI Sweden, Science for Life Laboratories, Stockholm, following standard methods. For the SCCs, the library preparation was performed by a semiautomatic NeoPrep station using the Illumina TruSeq Nano Kit (350 bp average insert size) and 25 ng of DNA starting material. The libraries of the bulk DNA samples were prepared with Illumina TruSeq PCR-free library preparations (350 bp average insert size). Sequencing was performed on Illumina HiSeq X, PE 2×150 bp.

**SC purity of cultured CD56$^+$ populations**. The purity of the SC populations isolated with our protocol was compared in different muscle biopsies by assessing the transcript levels of the SC marker Pax7. In parallel with single-cell sorting, CD56$^+$ cells were sorted in a 24-well plate (Supplementary Table 1) and cultured for 1 week to expand the population prior to RNA extraction. In addition, the purity of the single-cell culture was tested by assessing the percentage of clones derived from single-cell sorted cells that expressed the myocyte marker MyoD. RNA levels were tested in those SCCs that were able to grow confluent in a 24-well plate and 97.4% of clones (CES1 $n = 3$, CES6 $n = 9$, CES7 $n = 14$, and CES8 $n = 13$) were found positive. The relative expression of SC markers was measured using the same calibrator, the CD56$^+$ sorted population from CES3, expanded in culture for 4 weeks (Pax7 expression characterized in Supplementary Fig. 1b). As negative controls, a human fibroblast population and two clones from human subcutaneous fat were used. Negative control samples never showed amplification of the SC genes.

To further characterize the contaminant cell types that were present in our CD56$^+$ sorted populations, we expanded for 3 weeks and froze the unsorted populations left from the original cell sorting of the CES biopsies. Thawed cells were sorted for CD56 and fluorescence-activated cell sorting analysis was performed to assess the percentage of cells expressing markers of SCs (Pax7 monoclonal, Hybridoma Bank, 10 µg/ml) and bone marrow-derived cells (CD45-APC, clone HI30, BD Biosciences, USA). At least 7000 cells per sample were counted. In addition, 5000 cells were cytospun on glass slides and immunofluorescently labeled for the fibroblast marker TE-7 (CBL271, Millipore, CA, USA). At least 300 cells per sample were counted. Thawed cells sorted for CD56 were grown as single-cell clones with the same protocol used in the original experiment and again the myogenic origin was tested by MyoD expression. 100% of clones resulted positive (CES3 = 4, CES6 = 3, CES7 = 6, CES8 = 6).

**Verification of SC origin of the sequenced SCCs**. Culture of SCCs from CD56$^+$ sorted cells does not guarantee the exclusive growth of SC colonies, due to the presence of contaminant cells of myogenic and non-myogenic origin in the CD56$^+$ pool (see Supplementary Table 1). For this reason, the SC origin of the sequenced SCCs was tested in a myotube-formation assay or by assessing the expression of the SC markers Pax7, MyoD, and myogenin via qPCR. This test could be performed for 21 of the 29 clones. The remaining eight clones showed a scarce growth, and after DNA extraction, no cells were available for additional tests.

**Cell-culture assays**. Proliferation assays were performed on the seven biopsies included in the study (Supplementary Table 1). For every biopsy, at least 96 SCs were plated as single cells (CES1, CES7, and CES8: $N = 96$; CES2, CES3, CES4, and CES6: $N = 144$). After 2 weeks in culture, the percentage of cells able to form a colony of at least 16 cells was scored (colony-forming SCs). SCs that were able to grow up to 30,000–100,000 cells (confluent well of six-well plate) were scored positive for long-lasting proliferative potential independent of the time in culture. The differentiation assay was performed on confluent cells, by replacing growing medium with differentiating medium (DMEM-F12, 1% FBS) for 5 days. The clones were scored for the presence of visible myotubular structures and the RNA was extracted to verify the myogenic origin. For this assay, the original plating of the single-cell clones was used for biopsies CES1 $n = 37$, CES2 $n = 39$, and CES4 $n = 29$. For biopsies CES3, CES6, CES7, and CES8, the frozen expanded populations of unsorted cells were thawed and grown for 1 week, than sorted for CD56, single cell plated, and grown to confluency in 24-well plates (CES3 $n = 21$, CES6 $n = 37$, CES7 $n = 24$, and CES8 $n = 28$ SCCs).

**Long-culture SCCs**. Freshly isolated SCs from a young individual (Supplementary Table 1) were expanded in culture for 50 days using growth medium (DMEM F12 containing 1% ABAM and 20% FBS), then the SCs were single cell plated and cultured according to our protocol for clonal expansion of SCs from human SkMs. The clones and blood bulk DNA from the donor were sequenced, and the somatic variants were analyzed as a control for the in vitro-induced mutations.

**Somatic variant calling**. Raw reads were aligned to the human reference genome (GRCh37/hg19 assembly version), using bwa mem 0.7.12[41]. Samtools 0.1.19[42] was used for alignment sorting and indexing, and qualimap v2.2[43] for the alignment quality-control statistics. The raw alignments were then processed using the GATK best practice[44] with version 3.3 of the GATK software suite. GATK RealignerTargetCreator and IndelRealigner were used to realign around indels, Picard MarkDuplicates 1.120 to mark duplicates, and GATK Base-Recalibrator to recalibrate base quality scores. Finally, genomic VCF files were created using the GATK HaplotypeCaller 3.3. Reference files came from the GATK 2.8 resource bundle and steps were coordinated using Piper v1.4.0 (www.github.com/NationalGenomicsInfrastructure/piper).

To identify somatic variants, a specific pipeline was developed. For each SCC, the union of variants called with HaplotypeCaller (GATK)[45], MuTect2 (GATK 3.5.0), and FermiKit version r178[46] were subjected to further filtering steps. First,

the variants present in any of the SCCs were gathered in a comprehensive list of interesting positions, which was specific to each individual. For every clone of the individual, the read distribution of the interesting positions was derived from the .bam files and matched to the relative blood bulk. Variants were called when all of the following criteria were met: the read fraction supporting the alternative allele fell in the desired region (0.4–0.6), the read fraction in the blood was low (alternative < 0.1), and the coverage in both the clone and blood was at least 15×. Chromosomes X and Y were excluded from the analyses. Additional quality filters were applied as follows: the reads supporting the variants were on both strands, the maximum coverage was 1000×, and the variants that were located in problematic regions[47,48] and those detected in more than one individual were removed. The variants were annotated using the Ensembl Variant Effector Predictor from ref. [49]. Filtered somatic mutations in young, old, and long-term cultured human SC identified in the study are available in Supplementary Data 1–3.

**Estimate of false negative rate**. The estimate of false negative rate was done by counting how often heterozygous SNVs were detected in the clone bam file with an allele frequency that allows our pipeline to classify it as have occurred in vivo, i.e., 0.4–0.6 allele frequency. First, a set of high-confidence SNVs for which the individual was germline heterozygote (bulk and blood) and present with an allele frequency > 0.3 in the SweGen population was generated. Second, the germline heterozygote SNVs with a clone depth minimum of 15× were selected. Third, the number of SNVs that were detected in the clone bam file by GATK and/or Fermikit and/or MuTect2 were collected. Finally, the number of SNVs with allele frequency 0.4–0.6 were counted using the Samtools mpileup. Similarly, we estimated how often homozygous SNVs are detected with minor allele frequency under 0.1 in blood. High-confident homozygous SNPs were created in the same way as above, and Samtools mpileup was used to count the number of SNVs with allele frequency below 0.1 in the blood. To yield the false negative rate, the positive prediction rate (e.g., the fraction of SNVs with allele frequency 0.4–0.6 of the total number of SNVs and correct called homozygous SNVs) was subtracted from 1.

**Variant validation**. The variant validation was performed on technical replicates of WGS. Clone P2703_113 was sequenced twice with independent library preparations. Clone P2703_116 was split into 2 wells during the cell culture (1000 cell-stage) and resulted in 2 independently grown clones (P2703_116 and P2703_119) derived from the same ancestor cell. The DNA was extracted and sequenced independently, but clone P2703_119 was not included in the study. Variants were called in clones P2703_113 and P2703_116 (discovery set) according to our somatic variant calling pipeline. Called variants that had a minimum coverage of 10× in both the discovery and the validation sets were used for the validation. In total, 798 SNVs and 82 indels of clone P2703_113 and 1140 SNVs and 66 indels of clone P2703_116 were tested. Variants were considered validated when at least three reads supporting the alternative alleles were present in the validation set. As a control for the background signal, we validated the variants in unrelated clones, e.g., clones derived from a different founder cell obtained from the same or a different biopsy. For a WGS-independent validation, we used Agena genotyping (sequenom maldi-tof technology), which was performed at the Mutation Analysis Facility at the Karolinska Institute. The SNVs predicted to have high-impact consequences on the encoded protein were selected from all clones and tested when assays were designable ($n = 11$) in all clone, muscle bulk, and blood bulk DNA included in the study. The variants were considered validated when the expected genotype was found in the clone from which the SNV was discovered, and the reference allele was found in all the other samples, including the relative blood bulk sample. In addition, six SNVs selected based on their impact on muscle physiology were validated by digital droplet PCR (ddPCR; Bio-Rad) in the clone from which the SNV was discovered and the relative bulk samples.

**RNA extraction and qPCR**. RNA was extracted from muscle biopsies using Tri-Zol® Reagent (Invitrogen) and from cultured cells using the RNeasy Mini Kit (Qiagen), according to the manufacturers' instructions. cDNA synthesis was performed using random hexamers and SuperScript Reverse Transcriptase (Invitrogen). Quantitative reverse transcriptase PCR (RT-PCR) was performed using Taqman qPCR reagents and an ABI7500 fast system sequencing detection instrument (Applied Biosystems). Primers and probes were assays on demand from Applied Biosystem (Pax7 Hs00242962_m1; MyoD1, Hs00159528_m1; myogenin Hs01072232_m1) and the normalizer gene was glyceraldehyde 3-phosphate dehydrogenase (#4352665, Applied Biosystems).

**Digital droplet PCR**. The rare event detection (RED) analysis was performed using the QX200 ddPCR system (Bio-Rad, Hercules, CA, USA). DdPCR assays (Bio-Rad, Hercules, CA, USA) were designed to detect six SNVs, namely, *HSPG2* (chr1: 22174499G>A, assay id: dHsaMDS988410713), *TTN* (chr2: 179560728G>A, assay id: dHsaMDS896792909), *FN1* (chr2: 216272892C>T, assay id: dHsaMDS668887829), *FNDC1* (chr6: 159682281T>A, assay id: dHsaMDS711365861), *RAP1GDS1* (chr4: 99338610C>T, assay id:

dHsaMDS467491822), and *CACNA2D2* (chr3: 50426879C>T, assay id: dHsaMDS892017893). The PCR reactions were performed using 2× ddPCR Supermix for Probes (no dUTP) (Bio-Rad), 20× ddPCR mut assays (FAM/HEX labeled) (Bio-Rad), 5 U of *Hin*dIII restriction enzyme (New England BioLabs, Ipswich, MA, USA), and template DNA and ran according to the manufacturer's instructions. gBlocks gene fragments (IDT, Coralville, IA, USA), carrying one of the six SNVs each, were spiked into the human DNA sample and used for testing the ddPCR assays and optimizing the PCR thermal cycling conditions prior to the analysis of the samples. Approximately 30 ng of sample DNA per well were used in the analysis, except for the case of DNA from the additional clones from individual CES6 for which only 5 ng per well were used due to the low amount of available DNA. All samples were run in two or more replicate wells, and the data were merged from all the replicate wells to calculate the fractional abundances of the mutant alleles. Sample data were only accepted when falling within established detection parameters, which include a minimum of 3 positive droplets per sample and 10,000 accepted droplets per well.

Owing to the exonic position of the *HSPG2* and *FN1* mutations, the same assay could be used with both DNA and cDNA. To perform the RED on the cDNA samples, 1 μg of RNA was treated with DNase using RQ1 RNase-Free DNase (Promega, Madison, WI, USA) prior to the cDNA synthesis via the SuperScript First-Strand Synthesis System for RT-PCR (Invitrogen, Carlsbad, CA, USA), which was performed according to the manufacturer's instructions. The RED was performed as described for DNA. Approximately 50 ng of sample cDNA per well were used for the analysis, except for the case of cDNA from the SC population from a control individual, for which the amount of cDNA had to be reduced to 7.5 ng per wall to prevent overloading of the assay.

**Mutational signatures**. All analyses of base substitutions (Supplementary Fig. 2b) and mutational signatures (Fig. 1 and Supplementary Figs. 3 and 4) were carried out using the R-package MutationalPatterns[16,50] following the creators' instructions. In brief, base substitution patterns of the somatic mutations detected in SCCs were analyzed and visualized with MutationalPatterns using the reference genome set to "BSgenome.Hsapiens.UCSC.hg19". Using the function "mut_matrix", the relative contribution of the six single base substitution subtypes: C:G>A:T, C:G>G:C, C:G>T:A, T:A>A:T, T:A>C:G, and T:A>G:C, was obtained by including the sequence context 5′ and 3′ of each mutated base to obtain a 96 trinucleotide substitution count matrix. We performed de novo extraction of mutational signatures from the SCC mutations using non-negative matrix factorization[51] by applying the function "extract_signatures" to our data. We used the 96 trinucleotide substitution count matrices for all SCCs ($n = 29$) as input and specified that three signatures should be extracted (rank = 3). The contribution of the 96 possible trinucleotide substitution types to the extracted signatures was visualized using the function "plot_96_profile" (Supplementary Fig. 1f). The relative contribution of the extracted signatures to the mutational catalog was used for the PCA (Supplementary Fig. 1g). We used the function "fit_to_signatures" to assess how well the mutational catalog of our SCCs can be expressed as a combination of the previously identified cancer signatures (http://cancer.sanger.ac.uk/cosmic/signatures). Whole-genome data were then expressed as SNVs/Gbase/SCC and replotted (Fig. 1f). For each signature, the number of mutations per SCC was plotted and the linear fit with donor age was used to calculate the year increase (Fig. 1g). Analyses in R were done using RStudio Version 0.98.1025. Signature data were analyzed using the following pipeline to obtain principal components of the 29 sequenced SSCs and previously published sequence data from human fibroblasts[27]: (i) data were normalized on a per clone base; (ii) normalized data were transformed into principal components (NB: these principal components are the ones shown in the plots); (iii) 100,000 permutations of *k*-means clustering algorithm (with $k = 2$) have been performed on data restricted to the first three principal components. This pipeline has been implemented using MATLAB, and all programs have been run with default parameters.

**Genomic distribution of mutations**. The distribution of somatic mutations across the human genome (Supplementary Fig. 5a) was visualized using the function "plot_rainfall" of the R-package MutationalPatterns[16,50]. The observed and expected numbers of mutations in the different genomic regions (Fig. 2b, c) were analyzed using the function "genomic_distribution". Input data were the somatic mutations detected in SCCs, genomic regions with enough sequencing depth (15×) in every SCC, and the genomic regions that should be tested for enrichment or depletion of somatic mutations, all represented as GRanges objects[52]. Regulatory regions were extracted from Ensembl using the BioMart database "regulation", dataset "hsapiens_regulatory_feature" and GRCh "37". Promoter regions were extracted using the filter "regulatory_feature_type_name" set to "Promoter". Promoter flanking regions were extracted using the filter "regulatory_feature_type_name" set to "Promoter Flanking Region". Open chromatin regions were extracted using the filter "regulatory_feature_type_name" set to "Open chromatin". Transcription factor (TF)-binding regions were extracted using the filter "regulatory_feature_type_name" set to "TF binding site". To test for significant depletion or enrichment of somatic mutations in genomic regions, we used the function "enrichment_depletion_test" with the observed and expected number of mutations in the genomic regions as input. The results were visualized using the fuction "plot_enrichment_depletion", with the significance test data as input. Exon

data were extracted from Ensembl using the BioMart database "ensembl", dataset "hsapiens_gene_ensembl" and GRCh "37". Information on genes that were detected or not detected on RNA level in the SkM was extracted from the human protein atlas (http://proteinatlas.com). A list of genes detected in the SkM was obtained by concatenating the results from the search terms "tissue_specificity_rna:skeletal muscle;elevated AND sort_by:tissue specific score", "tissue_specificity_rna:any;expressed in all", and "tissue_specificity_rna:skeletal muscle;mixed" at http://proteinatlas.org. A list of genes not detected in the SkM was generated by the search term "tissue_specificity_rna:skeletal muscle; not detected" at http://proteinatlas.org. Based on Ensembl ID, we selected the subsets of all exons extracted from the BioMart database "ensemble" that were present in the list of genes detected in the SkM or in the list of genes not detected in the SkM.

The distribution of mutations in 173 muscle disease genes was tested against a null model as follows. SCCs were assigned to the young ($n = 13$) and old ($n = 16$) groups and all genes harboring a mutation were registered for each group. The total number of mutations with encode annotation to a gene (e.g., harbored in extended gene regions including exons, introns, and 5 kb upstream and downstream of each gene) was counted for each group. We sampled $10^4$ times the same number of exons from the list of all exons ("hsapiens_gene_ensembl" previously described) proportionally to their length and then computed how many extracted extended gene regions were present in the list of muscle disease genes. *P*-value and *Z*-score were then evaluated by comparing the empirical number of mutations with the distribution of the null-models using one-sided *t*-test. Full gene length was obtained from Gencode, release 19.

**Mutation load in specific genomic regions and gene sets**. For the analysis of genes, promoters and enhancers actively expressed during myoblast to myotube differentiation (Fig. 3a) data were downloaded from the FANTOM5 project at http://fantom.gsc.riken.jp/data/ (human_permissive_enhancers_phase_1_and_2_expression_tpm_matrix.txt.gz and hg19.cage_peak_phase1and2combined_counts_ann.osc.txt.gz)[53]. Expression data were available at 8 different time points during the myoblast to myotube differentiation (CNhs13847-14585): day 00 (basal), day 01, 02, 03, 04, 06, 08, 10, and 12 upon exposure to differentiation stimuli. Exploiting different time points, two expression sets, "basal" and "upon differentiation", were created. "Basal" corresponds to transcribed regions on day 00. For "upon differentiation", transcribed regions at all time points were intersected with expressed regions on day 00 and the non-overlapping portion of transcribed regions at all time points was taken. To determine actively transcribed promoters and enhancers, we determined an arbitrary threshold based on surveyed region size (see Supplementary Fig. 8). For enhancers, all regions with an expression score ≥1 were included and the considered regions were enlarged +/−100 bp. For promoters, all regions with an expression score ≥30 were included and the considered regions were enlarged of an asymmetric window of 860 bp upstream and 100 bp downstream. For exons, gene names were derived from expressed regions reported in the promoter list and the exonic regions of each gene were derived from "hsapiens_gene_ensembl" previously described. Three different donor replicates were used and regions transcribed with intensity equal or above the threshold in at least one replicate were included. Size of surveyed regions was 4525728 and 32409968 bp for enhancers in "basal" and "upon differentiation", respectively, 5335985 and 3903736 for promoters, and 17542615 and 11123995 for exons. Somatic SNVs and indels were annotated to the described regions and quantified for every SCC and the average of mutations in young and old SCCs was reported. When enhancer and promoter regions overlapped, mutations were attributed to either one based on manual investigation of the region.

For the analysis of the numbers of mutations in gene sets involved in muscle function (Fig. 4b), genes were derived as described in Supplementary Tables 7 to 9. Lists of exonic mutations in young and old clones were screened and every mutation that was annotated with a gene name included in the gene set was scored. The average number of mutations per gene was calculated by dividing the number of mutations in the gene set by the number of tested SCC (young $n = 13$, old $n = 16$) and the number of genes in the gene set.

**Statistical analyses**. Unless otherwise indicated, *P*-values were calculated using two-tailed distribution, two-sample unequal variance Student's *t*-tests, with significance defined as $P < 0.05$ (*$P < 0.05$, **$P < 0.01$, ***$P < 0.001$). The results are presented as the mean ± standard error of the mean (SEM). All calculations were performed using the GraphPad Prism software. The linear fits between mutation numbers and age shown in Fig. 1 were obtained using a robust mixed model where the dependent variable is the number of mutations or a given mutational signature, the fixed effect is age, and the random effect is the individual. Bonferroni correction for the number of tested signatures was applied in Fig. 1f. Analyses were performed in R.

**Data availability**. The sequence data that support the findings of this study are available within the article and its Supplementary Information (Supplementary Data 1–3). Raw sequence data are only available from the corresponding author upon request and will be made available to research projects that fulfill the

informed consent, owing to regulations pertaining to the authors' deposition of these data in public repositories.

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

## Acknowledgements

We acknowledge the healthy donors who contributed tissue samples for the study. Authors wish to thank Carsten Daub for critical discussion on FANTOM data and Nils-Göran Larsson for comments on the manuscript. This study was supported by grants to M.E. from the Swedish Research Council and the Center for Innovative Medicine; to I.F. from the Hagelen and Osterman Foundations and Svenska Läkaresällskapet; to P.V. from the Osterman and Stohnes Foundations; to A.P. and C.B. from Marie Skłodowska-Curie, grant agreement No. 734439 (INFERNET); and to T.G. from the Swedish Medical Research Council (2013-09305) and the Marcus and Marianne Wallenberg foundation. A.J. and M.L. are supported by a grant from the Wallenberg Foundation to the Wallenberg Advanced Bioinformatics Infrastructure. The authors acknowledge support from Science for Life Laboratory, the Knut and Alice Wallenberg Foundation, the National Genomics Infrastructure funded by the Swedish Research Council, and Uppsala Multi-disciplinary Center for Advanced Computational Science for assistance with massively parallel sequencing and access to the UPPMAX computational infrastructure. We thank the Mutation Analysis Core Facility (MAF) at the Karolinska University Hospital for their support to this work. This study was performed in part at the Live Cell Imaging

Unit/Nikon Center of Excellence, Department of Biosciences and Nutrition, Karolinska Institutet, Huddinge, Sweden.

## Author contributions

I.F., H.F. and M.E. designed the study. I.F., K.O., P.V., T.G. and H.F. collected samples. I.F., P.V., G.R. and H.F. performed experimental work. I.F., A.J., P.V., P.L., H.T.H., M.L., G.R., C.B., A.P., P.P., T.G., H.F. and M.E. analyzed and interpreted data. I.F., H.T.H., C.B., A.P. and P.P. performed statistical analysis. I.F. wrote the first draft of the manuscript and all authors critically revised the manuscript. I.F. and M.E. obtained the funding.

## Additional information

**Competing interests:** The authors declare no competing financial interests.

