## [Peer Review File · Nature Communications]

Reviewer #1 (Remarks to the Author):

In this work, Franco et al study the incidence of mutations occurring in human skeletal muscle satellite cells throughout life. To this end they compare the nature and the frequency of mutations found in satellite cells from three young men (20-24 year old) and four old men (70-78 year old). Their work involves the isolation of clones of satellite cells from muscle biopsies, their expansion for Whole Genome Sequencing and the identification of somatic mutations, Single Nucleotide Variants, SNVs, and INDELS.

Their approach requires a phase of expansion in culture in order to have access to sufficient amounts of DNA. During this phase the cells undergo about twenty divisions. They exclude that these SNVs could be of germ-line origin since they are absent from the DNA of whole blood and bulk muscle. Evidence is also provided that the profile of SNVs found in clones of satellite cells differ from the one normally seen as culture induced.

They calculate a yearly mutation rate for satellite cells and hypothesize that satellite cells, although viewed as quiescent, undergo an average of 5 divisions per year. This does not contradict previous observations in animal models in which a small percentage of satellite cells were shown to go through DNA synthesis in adults, (3.5% of them after a nine days labeling period in 4 month old mice). It is widely accepted that quiescence of adult satellite cells does not exclude a small fraction of cells could undergo proliferation. Moreover this is also influenced by the life style of individuals.

Their findings can be summarized as follows:

The absolute number of SNVs increases with age. This not surprising.

The rate of accumulation of mutations is higher in exons and promoter regions of old satellite cells than in introns. This is intriguing; it is proposed that this could result from a weakening of the protective mechanisms in old cells. This leads the authors to hypothesize that old satellite cells maybe less functional for muscle repair. This is an interesting piece of work that sheds light on genomic alterations as an intrinsic factor potentially affecting satellite functionality in the course of ageing. Some ex vivo evidence of this loss of functionality is provided (fig 3B and C), however it is not clear if this ex vivo assay is significant (see fig 3C, is 65% significantly different from 80%?). Thus the evidence sounds circumstantial at this stage.

The analyses presented in Figure 4 sounds much more like assumptions than demonstrations. The discussion of this paper is too limited and exclusively based on the data and hypotheses presented in this work. In particular, there is no mention of the involvement of extrinsic factors. Extrinsic factors have been involved in the loss of functionality of satellite cells upon ageing, such as the satellite cell

niche, their environment, interstitial cells, secreted growth factors acting through local or systemic pathways.

The style adopted by the authors sounds very specialized and thus may not be adapted to a large audience. This is my feeling as a cell biologist working in the field of developmental and regenerative myogenesis.

Overall this work appears more descriptive than demonstrative and the evidence for a loss of functionality due to the accumulation of mutations in satellite cells sounds circumstantial at this stage.

Reviewer #2 (Remarks to the Author):

In this manuscript, Franco et al. examine somatic mutagenesis in satellite cells by performing WGS sequencing of expanded single cells taken from seven individuals at different ages. The analysis of the data reveals somatic mutations from different mutational signatures, accumulation of somatic mutations with age, association between high mutational burden and differentiation defects, and propagation of missense mutations from the satellite cells to the muscle.

Overall, this is a fascinating study and one of the first to demonstrate the importance of examining age-associated accumulation of somatic mutations in normal tissue and the degradation processes of the human body. This manuscript will be of interest to most researchers and/or clinicians performing biomedical research. However, I do have several concerns about the methods/analysis used in this manuscript:

Mutational signatures: The authors performed de novo extraction that reveals three mutational signatures. From the de novo extraction, one can clearly see the presence of COSMIC signatures 1, 5, and 18 but not much else. In contrast, they report in the main manuscript many more COSMIC signatures in these samples. Some of these signatures are highly unlikely to be present. For example, signature 29 is due to tobacco chewing and it have been only found in oral cancers from India. Another example will be signature 16, signature 16 has been found only in liver cancer/tissue and it exhibits a strong transcriptional strand bias with damage almost exclusively on adenine. Similar arguments can be made for signatures 3 (HR deficiency), 26 (MSI signatures), 30 (BER deficiency), etc. It seems that the authors are overfitting the COSMIC signatures to their data. Unfortunately, this

is a common problem with existing packages for analysis of mutational signatures. I suggest that the authors evaluate whether adding any mutational signature beyond 1, 5 and 18 (which they find in the de novo extraction) improves significantly their solution accuracy (usually, measured with cosine similarity). The mutational signatures analysis also has implications for the subsequent analysis.

Correlation with age: The correlation with age was performed using linear mixed model. Unfortunately, linear mixed models are affected by outliers especially for small number of samples. The authors should instead use robust mixed models for their analysis. Also, the main text refers to p-values but from the methods these seem to be q-values (i.e., FDR corrected p-values).

Mutation depletion: The authors report that “mutation depletion in young SCCs was stronger than in old SCCs..., suggesting that the mechanisms protecting the functional regions weakened with aging.” I do not completely understand this argument. The mutations in the expressed part of the genome are preferentially repaired by TC-NER, which in turn has certain level of fidelity. For example, let’s imagine that TC-NER misses one damage per year leading to a mutation in the transcribed region. In a twenty-four year old person this will be 24 extra mutations, while in seventy-five year old this will be 75 extra mutations. In this case, TC-NER has exactly the same fidelity (its function does not weaken with aging) but one would observe a mutation depletion in young people compared to old people.

In summary, I find this manuscript to be very interesting. However, the authors must address the outlined limitations in their analysis before I can recommend it for acceptance.

**Point-by-point response to the reviewers comments
NCOMMS-17-25674; Franco et al.**

Reviewers' comments:

Reviewer #1 (Remarks to the Author):

In this work, Franco et al study the incidence of mutations occurring in human skeletal muscle satellite cells throughout life. To this end they compare the nature and the frequency of mutations found in satellite cells from three young men (20-24 year old) and four old men (70-78 year old). Their work involves the isolation of clones of satellite cells from muscle biopsies, their expansion for Whole Genome Sequencing and the identification of somatic mutations, Single Nucleotide Variants, SNVs, and INDELS.

Their approach requires a phase of expansion in culture in order to have access to sufficient amounts of DNA. During this phase the cells undergo about twenty divisions. They exclude that these SNVs could be of germ-line origin since they are absent from the DNA of whole blood and bulk muscle. Evidence is also provided that the profile of SNVs found in clones of satellite cells differ from the one normally seen as culture induced.

They calculate a yearly mutation rate for satellite cells and hypothesize that satellite cells, although viewed as quiescent, undergo an average of 5 divisions per year. This does not contradict previous observations in animal models in which a small percentage of satellite cells were shown to go through DNA synthesis in adults, (3.5% of them after a nine days labeling period in 4 month old mice). It is widely accepted that quiescence of adult satellite cells does not exclude a small fraction of cells could undergo proliferation. Moreover this is also influenced by the life style of individuals.

Authors reply: We apologize for the omission of reference to previous work. We have in this revised version included the citation of the work mentioned by the reviewer in the discussion. The sentence now reads: “Moreover, we cannot establish whether the proliferation rate is attributable to a small fraction of dividing SCs, as shown in the muscle of adult mice³¹”. Ref 31: Schultz, E., Gibson, M.C. & Champion, T. Satellite cells are mitotically quiescent in mature mouse muscle: an EM and radioautographic study. *J Exp Zool* 206, 451-456 (1978).

Their findings can be summarized as follows:

The absolute number of SNVs increases with age. This not surprising.

The rate of accumulation of mutations is higher in exons and promoter regions of old satellite cells than in introns. This is intriguing; it is proposed that this could result from a weakening of the protective mechanisms in old cells. This leads the authors to hypothesize that old satellite cells maybe less functional for muscle repair. This is an interesting piece of work that sheds light on genomic alterations as an intrinsic factor potentially affecting satellite functionality in the course of ageing. Some ex vivo evidence of this loss of functionality is provided (fig 3B and C), however it is not clear if this ex vivo assay is significant (see fig 3C, is 65% significantly different from 80%?). Thus the

evidence sounds circumstantial at this stage.

Authors reply: We apologize for the unclear presentation of the data. We have in this revised version of the manuscript changed the format of the graph in fig 3B to express the percentage of differentiating SCCs as a function of the average number of somatic mutations in each individual. This shows that the higher the number of somatic mutations the lower the ability to differentiate and support the data presented in fig 3C where the number of cells able to proliferate negatively correlated with the average number of mutations per biopsy. Both graphs are significant. These analyses, together with figures 3D-F where, thanks to the data available from the clonal culture of the sequenced SCCs we had the unique opportunity of analyzing whole genome data and *ex vivo* proliferation data from 29 single satellite cells, show that mutation burden is a factor contributing to the loss of functionality in satellite cells.

The analyses presented in Figure 4 sounds much more like assumptions than demonstrations.

Authors reply: We disagree with the reviewer. The analyses in Figure 4 are a quantification of the genomic alterations occurring with aging in genes recognized as crucial for muscle function. The analyses in figure 3 and 4 together prove a functional effect of the somatic mutations.

The discussion of this paper is too limited and exclusively based on the data and hypotheses presented in this work. In particular, there is no mention of the involvement of extrinsic factors. Extrinsic factors have been involved in the loss of functionality of satellite cells upon ageing, such as the satellite cell niche, their environment, interstitial cells, secreted growth factors acting through local or systemic pathways.

Authors reply: We appreciate the comment by the reviewer. In this revised version of the manuscript the discussion of the paper has been significantly extended, see pages 8-11. A new paragraph addressing the importance of intrinsic and extrinsic factors in driving the aging of the skeletal muscle has been added (page 10, third paragraph): “We wondered whether the higher mutation burden could cause an impairment of the satellite cell functions, e.g. proliferation and differentiation. Alterations in the stem cell function during aging have been ascribed to deregulation of both extrinsic and intrinsic factors³⁵. Extrinsic factors include molecules systemically delivered to the muscle or secreted by the local niche in the microenvironment where the SCs reside. It has been demonstrated that the SC environment changes over time and influences the SC self-renewal and capacity for tissue regeneration³⁶. On the other hand, the age-related reduction in the number of SCs that are able to repopulate the niche and differentiate into muscle fibers has been demonstrated to also rely on cell-intrinsic factors, such as the activation of signalling pathways^{37,38} and epigenetic mechanisms³⁹. With our model we could explore how an intrinsic factor, i.e. genetic alterations, affects the SC activity, providing unique genome-wide data and direct assessment of the *ex vivo* performance of SCs of different ages....”

The style adopted by the authors sounds very specialized and thus may not be adapted to a large audience. This is my feeling as a cell biologist working in the field of developmental and regenerative myogenesis.

Authors reply: We appreciate the comment from the reviewer. We have in this revised version extended the introduction to include additional background on somatic mutations, (page 2, second and third paragraph). Moreover, the discussion has been extended and adapted to a less specialized audience, see page 8-11.

Overall this work appears more descriptive than demonstrative and the evidence for a loss of functionality due to the accumulation of mutations in satellite cells sounds circumstantial at this stage.

Authors reply: We appreciate the comment from the reviewer but a direct demonstration is technically challenging. As discussed in the first and second point of this rebuttal, we like to emphasize that this paper provides unforeseen data on somatic mutations and the proliferation ability of 29 human single satellite cells of different ages. Our analyses point to an effect of the somatic mutation burden in reducing the satellite cell fitness in aged individuals and this is in line with impaired muscle function with aging.

Reviewer #2 (Remarks to the Author):

In this manuscript, Franco et al. examine somatic mutagenesis in satellite cells by performing WGS sequencing of expanded single cells taken from seven individuals at different ages. The analysis of the data reveals somatic mutations from different mutational signatures, accumulation of somatic mutations with age, association between high mutational burden and differentiation defects, and propagation of missense mutations from the satellite cells to the muscle.

Overall, this is a fascinating study and one of the first to demonstrate the importance of examining age-associated accumulation of somatic mutations in normal tissue and the degradation processes of the human body. This manuscript will be of interest to most researchers and/or clinicians performing biomedical research. However, I do have several concerns about the methods/analysis used in this manuscript:

Authors reply: We thank the reviewer for the nice comments of our work. We have tried to answer the concerns and performed the additional analysis as suggested. The revised version of the manuscript has been updated accordingly.

Mutational signatures: The authors performed de novo extraction that reveals three

mutational signatures. From the de novo extraction, one can clearly see the presence of COSMIC signatures 1, 5, and 18 but not much else. In contrast, they report in the main manuscript many more COSMIC signatures in these samples. Some of these signatures are highly unlikely to be present. For example, signature 29 is due to tobacco chewing and it have been only found in oral cancers from India. Another example will be signature 16, signature 16 has been found only in liver cancer/tissue and it exhibits a strong transcriptional strand bias with damage almost exclusively on adenine. Similar arguments can be made for signatures 3 (HR deficiency), 26 (MSI signatures), 30 (BER deficiency), etc.

Authors reply: We appreciate the suggestion by the reviewer. We have now done a complete analysis of cosine similarity between our signatures A,B,C (*de novo* extracted from the somatic mutation catalogues of 29 satellite cell clones) and COSMIC signatures. We found that signatures A,B,C are a mixture of signatures 5, 1, 8 and 16. While the cosine similarity with signature 18 is lower (Signatures A=0.46; B= 0.37; C=0.50). These results are now reported in the new Supplementary Fig. 4A.

Here are the top 3 results of cosine similarity between our de novo extractions and COSMIC signatures:

Signature A		
Signature.5	Signature.8	Signature.16
0.865	0.776	0.770

Signature B		
Signature.5	Signature.1	Signature.8
0.773	0.769	0.695

Signature C		
Signature.5	Signature.1	Signature.8
0.832	0.785	0.785

Reviewer #2 continue;

It seems that the authors are overfitting the COSMIC signatures to their data. Unfortunately, this is a common problem with existing packages for analysis of mutational signatures. I suggest that the authors evaluate whether adding any mutational signature beyond 1, 5 and 18 (which they find in the de novo extraction) improves significantly their solution accuracy (usually, measured with cosine similarity). The mutational signatures analysis also has implications for the subsequent analysis.

Authors reply: We appreciate the comment by the reviewer and we have now tried to fit our data to 3, instead of all COSMIC signatures. The best combination of 3 signatures is: 1, 5, 8. This is significantly higher than the one proposed by the referee: 1,5,18, and

explains 95% of data. Even though the combination 1,5,8,16, is more similar to the data presented in the first version of the paper, the cosine similarity of the fit with 1,5,8,16 is not significantly higher than the fit with 1,5,8 (see below). For this reason, and to avoid overfitting the data, we have in the revised version of the manuscript changed panels C,D,E,F,G of Figure 1 to only these 3 signatures. The modified analysis of data stresses the importance of signature 5 in the mutation processes shaping the genome of SCCs during aging. The manuscript text has been updated and extended accordingly.

p values:

2-de novo vs. 1,5,8 = 0.0002854872

3-de novo vs. 1,5,8 = 1.817267e-05

2-de novo vs 1,5,8,16 = 0.00101784

3-de novo vs 1,5,8,16 = 8.226704e-05

1,5 vs 1,5,8 = 0.0007431171

1,5 vs 1,5,8,16 = 0.000229841

1,5,8 vs 1,5,18 = 0.01670117

1,5,8 vs 1,5,8,16 = 0.7005253

1,5,8 vs COSMIC ALL = 0.1259912

1,5,18 vs 1,5,8,16 = 0.006449075

If we do the same complete analysis including the 4 long-culture SSCs we find that the new signature C can be mainly attributable to signature18 and is almost exclusively found in long-culture clones.

The presence of signature C/18 in long culture clones is in agreement with the occurrence of *in vitro*-derived mutations. For this reason we included a new supplementary figure 4B to show the contribution of signature 18 to long-culture SCC mutation catalogues and its very low contribution to regularly cultured clones. These data demonstrate that our regular protocol for SCC culturing produces very low levels of *in vitro*-occurred artifacts.

Reviewer #2 continue;

Correlation with age: The correlation with age was performed using linear mixed model. Unfortunately, linear mixed models are affected by outliers especially for small number of samples. The authors should instead use robust mixed models for their analysis.

Authors reply: We have in this revised version of the manuscript updated our analysis using the robust mixed models as suggested by the reviewer. New P-values have been included in Figure 1B and 1F.

Also, the main text refers to p-values but from the methods these seem to be q-values (i.e., FDR corrected p-values).

Authors reply: The reviewer is correct. However, we have in this revised version used Bonferroni correction for multiple testing (Fig. 1F) and kept the nomenclature P-value.

Mutation depletion: The authors report that “mutation depletion in young SCCs was stronger than in old SCCs..., suggesting that the mechanisms protecting the functional regions weakened with aging.” I do not completely understand this argument. The mutations in the expressed part of the genome are preferentially repair by TC-NER, which in turn has certain level of fidelity. For example, let’s imagine that TC-NER misses one damage per year leading to a mutation in the transcribed region. In a twenty-four year old person this will be 24 extra mutations, while in seventy-five year old this will be 75 extra mutations. In this case, TC-NER has exactly the same fidelity (its function does not weakened with aging) but one would observe a mutation depletion in young people compared to old people.

Authors reply: We appreciate the comment from the reviewer. The text has been rephrased and now reads: “Interestingly, both introns and exons are depleted of mutations, and this is in line with the fact that they are actively transcribed. However, only exons show a difference between young and old SCCs, with the young exons being more efficiently protected from mutations. This difference can also be observed in regulatory regions and can be attributable to either a progressive accumulation of mutations due to errors in the transcription-coupled repair or an age-related decay of the mechanisms in charge of specifically repairing the exons³³”.

In summary, I find this manuscript to be very interesting. However, the authors must address the outlined limitations in their analysis before I can recommend it for acceptance.

Reviewer #1 (Remarks to the Author):

The manuscript has been extensively rewritten, particularly the introduction and the conclusion that have been extended and enriched; this makes the manuscript much easier to read and more accessible to a wider audience. Moreover all my comments and criticisms were convincingly addressed. As I said earlier, this is an interesting piece of work that sheds light on genomic alterations as an intrinsic factor potentially affecting satellite cell functionality in the course of ageing.

Reviewer #2 (Remarks to the Author):

The authors have taken great care to address my previous concerns. I have no additional comments and I recommend the manuscript for acceptance.

REVIEWERS' COMMENTS:

Reviewer #1 (Remarks to the Author):

The manuscript has been extensively rewritten, particularly the introduction and the conclusion that have been extended and enriched; this makes the manuscript much easier to read and more accessible to a wider audience. Moreover all my comments and criticisms were convincingly addressed. As I said earlier, this is an interesting piece of work that sheds light on genomic alterations as an intrinsic factor potentially affecting satellite cell functionality in the course of ageing.

Authors reply: We appreciate the support of the reviewer.

Reviewer #2 (Remarks to the Author):

The authors have taken great care to address my previous concerns. I have no additional comments and I recommend the manuscript for acceptance.

Authors reply: We appreciate the support of the reviewer.